# Associations of Caregiver Cooking Skills with Child Dietary Behaviors and Weight Status: Results from the A-CHILD Study

**DOI:** 10.3390/nu13124549

**Published:** 2021-12-18

**Authors:** Yukako Tani, Aya Isumi, Satomi Doi, Takeo Fujiwara

**Affiliations:** 1Department of Global Health Promotion, Tokyo Medical and Dental University (TMDU), 1-5-45 Yushima, Bunkyo-ku, Tokyo 113-8519, Japan; tani.hlth@tmd.ac.jp (Y.T.); isumi.hlth@tmd.ac.jp (A.I.); doi.hlth@tmd.ac.jp (S.D.); 2Japan Society for the Promotion of Science, Tokyo 113-8519, Japan

**Keywords:** cooking skills, home cooking, vegetable intake, obesity, children

## Abstract

We examined whether caregiver cooking skills were associated with frequency of home cooking, child dietary behaviors, and child body weight status in Japan. We used cross-sectional data from the 2018 Adachi Child Health Impact of Living Difficulty study, targeting primary and junior high school students aged 9–14 years in Adachi City, Tokyo, Japan (*n* = 5257). Caregiver cooking skills were assessed using a scale with good validity and reliability modified for use in Japan. Child heights and weights derived from school heath checkup data were used to calculate WHO standard body mass index z-scores. After adjusting for potential confounders, caregivers with low-level cooking skills were 4.31 (95% confidence interval (CI): 2.68–6.94) times more likely to have lower frequency of home cooking than those with high level of cooking skills. Children with low-level caregiver cooking skills were 2.81 (95% CI: 2.06–3.84) times more likely to have lower frequency of vegetable intake and 1.74 (95% CI: 1.08–2.82) times more likely to be obese. A low level of caregiver cooking skills was associated with infrequent home cooking, unhealthy child dietary behaviors, and child obesity.

## 1. Introduction

In the last 50 years, people in developed countries have shifted toward meals away from home and cooking at home less [1,2,3]. In the United States, the percentage of daily energy consumed from home food sources dropped by approximately 25% from the 1960s to 2000 [1]. In Japan, household expenditure on precooked food increased by 26% from 1993 to 2015, and eating out is becoming more widespread among younger generations [4]. Alongside the decrease in home cooking, the idea that homemakers, especially women, should be educated to feed their families seems to have become outmoded [5]. However, the obesity epidemic has led to growing concerns about poor diets among children and increasing calls to reaffirm the importance of basic food preparation and cooking skills to prevent poor diets and chronic diet-related diseases [5].

Lifestyle changes in response to the coronavirus disease 2019 (COVID-19) pandemic suddenly increased the need for home cooking. In the United States before and during the initial peak of the COVID-19 pandemic, cooking meals at home increased from 4.49 to 5.18 days per week [6]. Canadian families with young children described that their greatest change since COVID-19 was spending more time cooking [7]. In China, even post-lockdown, 65% of people reported that they cooked more at home compared with the previous year [8]. Meanwhile, the COVID-19 pandemic is leading children toward unfavorable obesity-promoting behaviors, such as decreased physical activity, increased screen time, and greater consumption of snack foods [7,9]. Therefore, caregivers need to acquire the ability to create a healthy eating environment to prevent their child from having a poor diet and becoming obese. However, there is limited knowledge on the relationships between caregiver’s ability to prepare meals and their child’s diet and weight status.

Recently, evidence has been accumulating on the dietary benefits of home cooking. A systematic review confirmed dietary benefits of home cooking, including greater consumption of healthier food groups, enhanced healthy eating self-efficacy, and improved adherence to several healthy dietary recommendations [10]. Beyond dietary outcomes, a population-based study in the United Kingdom showed that more frequent consumption of home-cooked meals was associated with a greater likelihood of having normal weight and body fat status among adults [11]. Studies on Japanese children and adolescents showed that infrequent home cooking was associated with obesity, higher blood pressure, and lower high-density lipoprotein-cholesterol [12,13].

Cooking skills may be critical to encourage home cooking and improve the quality of meals [10]. Several studies reported that high level of cooking skills was associated with lower consumption of ready meals, convenience food, and ultra-processed food among adults [14,15,16]. Meanwhile, intervention studies demonstrated that improvement in cooking skills led to increased cooking confidence and consumption of vegetables and fruits [17,18]. A recent population-based study among older Japanese adults showed that a low level of cooking skills was associated with lower frequency of home cooking, vegetable/fruit intake, higher frequency of eating out, and underweight status [19]. However, most existing studies have focused on dietary benefits among adults, and limited research has examined the associations of caregiver cooking skills with child diet and weight status.

The aim of the present study was to examine the associations of caregiver cooking skills with frequency of home cooking, child dietary behaviors, and child body weight status.

## 2. Materials and Methods

### 2.1. Study Design and Subjects

The Adachi Child Health Impact of Living Difficulty (A-CHILD) study was established in 2015 to evaluate the determinants of health among children in Adachi City, Tokyo, Japan [20]. We used cross-sectional data from 2018 that covered caregivers and their children in three grades: fourth-grade, sixth-grade, and eighth-grade. The survey was conducted in all public elementary schools for fourth-grade children, nine selected elementary schools for sixth-grade children, and seven junior high schools for eighth-grade children [20]. In 2018, self-reported questionnaires were distributed to 6605 children (5311 fourth-grade, 618 sixth-grade, and 676 eighth-grade). Children and their caregivers completed the questionnaires at home and then returned the completed questionnaires to their school. A total of 5793 pairs (4605 fourth-grade, 556 sixth-grade, and 632 eighth-grade) of children and their caregivers returned the questionnaires (response rate: 88%). Of these, 5382 pairs (4290 fourth-grade, 514 sixth-grade, and 578 eighth-grade) provided informed consent, returned all questionnaires, and could be linked with health checkup data (consent rate: 93%). The present analysis was carried out using data for 5257 pairs, after the following exclusions for missing information: child age (*n* = 23); caregiver cooking skills and frequency of home cooking (*n* = 26); child month of birth, height, and weight (*n* = 10); and child dietary behaviors (frequency of vegetable intake and breakfast consumption) (*n* = 66). The A-CHILD protocol and use of the data for this study were approved by the Ethics Committee at Tokyo Medical and Dental University (No. M2016-284).

### 2.2. Frequency of Home Cooking

Frequency of home cooking was evaluated by caregivers using the following question: ‘How many times did you or someone else in your family cook meals for your children at home? Circle the answer that best applies for the past month’. A cooked meal was defined as a simple meal, such as a fried egg [12]. The five response items were: ‘almost every day’, ‘4–5 days/week’, ‘2–3 days/week’, ‘a few days/month’, and ‘rarely’. We defined <3 times a week as low frequency of home cooking because it was reported to be associated with child obesity and cardiovascular risk [12,13].

### 2.3. Child Body Weight Status

Child height and weight were measured in schools during health checkups by school teachers according to standardized protocols [21]. Height was measured to the nearest 0.1 cm using a portable stadiometer and weight to the nearest 0.1 kg on digital scales, without shoes and in light clothing. Body mass index (BMI) was calculated by dividing the weight (in kilograms) by the square of the height (in meters). BMI was expressed as a z-score representing the deviation in standard deviation units from the mean of a standard normal distribution of BMI specific to age and sex, according to the WHO Child Growth Standards. Child BMI was categorized as underweight (<−2SD), mild-underweight (−2SD to <−1SD), normal (−1SD to <+1SD), overweight (+1SD to <+2SD), and obese (≥+2SD) using standard deviation cut-off points [22].

### 2.4. Child Dietary Behaviors

Child frequency of vegetable intake was assessed by caregivers using the question ‘How often did your child eat vegetable dishes? Circle the answer that best applies for the past month’. The three response items were: ‘twice/day’, ‘once/day’, and ‘<3 times/week’. Respondents who ate vegetables and fruit less than once a day were categorized as having low frequency of vegetable intake. This cutoff point was defined by prevalence to the 10th percentile of the included children (Table 1). Child frequency of breakfast intake for the past month was assessed by self-reporting with responses of ‘every day’, ‘often’, and ‘rarely/never’, with ‘rarely/never’ defined as breakfast skipping.

### 2.5. Caregiver Cooking Skills

Caregiver cooking skills were assessed using a modified cooking skills scale designed with consideration of basic Japanese cooking methods and typical meals [19]. The scale consisted of five items: (1) able to peel fruits and vegetables; (2) able to make stir-fried meat and vegetables; (3) able to make miso soup; (4) able to make stewed dishes; and (5) like to cook. Items 1 to 4 reflected basic cooking methods and were adopted from the Japanese cooking skills score [19]. Item 5 was newly added for the present study as an indicator of cooking skills, because a previous study on life-course trajectories of cooking skills found that liking to cook was a characteristic of people who maintained a high level of cooking skills [23]. Participants were asked to evaluate their own cooking skills on a six-point scale ranging from ‘do not agree at all’ (=0) to ‘agree very much’ (=5). A high score meant that the caregiver had high confidence in their cooking skills. In psychometric testing, one factor with eigenvalue >1 was found and accounted for 92.3% of the variance. The Cronbach’s α for the cooking skills scale in the study sample was 0.78. Factor loadings ranged from 0.3 (item 5) to 0.9 (item 2). We calculated the mean scores of the five items and divided the results into two categories: high (score > 4.0) and low (score ≤ 4.0) as described previously [19].

### 2.6. Covariates

Child age, cohabitation status (parents, parents and grandparent(s), single parent and grandparent(s), single parent, or other), other children in household (yes or no), household annual income (<3.00, 3.00–5.99, 6.00–9.99, or ≥10.0 million Japanese yen), respondent (mother, father, or other), parental age (<35, 35–44, or ≥45 years), mother’s employment status (full-time, part-time, self-employed, side work, not employed, or other), and parental height and weight were assessed via the caregiver report. Parental BMI was calculated using self-reported height in centimeters and weight in kilograms. Standard categories of BMI were used to characterize parents as underweight (<18.5 kg/m^2^), normal (18.5–24.9 kg/m^2^), overweight (25.0–29.9 kg/m^2^), or obese (≥30.0 kg/m^2^) [24]. Participants with missing data on covariates were included in the analysis as dummy variables.

### 2.7. Statistical Analysis

First, participants were stratified by level of cooking skills, and differences between groups were analyzed using Pearson’s chi-square test. Second, multiple comparisons for the cooking skills scale were performed using a mixed linear model procedure to examine which cooking skills participants rated as difficult. The peeling scale used as a reference and the participant identification code was included as a random effect. Third, multiple comparisons between respondents (mother, father, and other) were analyzed using Dunnett’s pairwise comparison method with mother as the reference category. Fourth, we calculated adjusted odds ratios (ORs) with 95% confidence intervals (CIs) for low frequency of home cooking, child low frequency of vegetable intake, and child breakfast skipping using logistic regression. Fifth, we calculated adjusted relative risk ratios with 95% CIs for underweight, mild-underweight, overweight, and obese using multinomial logistic regression, with normal as the reference category. The models were adjusted for potential confounding factors that were associated with level of cooking skills in the first analysis. Finally, we conducted a mediation analysis to determine the proportion of the association between caregiver cooking skills and child weight status mediated by frequency of home cooking. We estimated the natural direct effects, controlled direct effects, and natural indirect effects of mediators after controlling for all covariates using the Paramed package in Stata [25]. The exposure was treated as a binary variable, with 0 representing a high level of caregiver cooking skills and 1 representing a low level of caregiver cooking skills. The mediators and outcomes were treated as continuous variables. Because the association between caregiver cooking skills and child weight status was U-shaped, the mild-underweight and underweight children were excluded from the mediation analysis of the relationship with child obesity. All analyses were conducted using Stata Version 15 (Stata Statistical Software; StataCorp LP, College Station, TX, USA).

## 3. Results

The characteristics of the children and caregivers are summarized in Table 1. Half of the children were girls, about 80% were fourth-grade and lived with their parents, 81% had siblings, and 11% had families with annual incomes below 3.00 million yen. A total of 10.8% ate vegetable dishes less than three times a week, 2.5% were breakfast skippers, 2.3% were underweight, and 5.5% were obese. The majority of the caregiver respondents were mothers (91%); 8% were fathers. The most common mother’s employment status was part-time (48%), followed by full-time (21%). A total of 2.6% of households cooked less than three times a week. Approximately 5% of caregivers (*n* = 247) were classified as having a low level of cooking skills. Children who had caregivers with a low level of cooking skills tended to live with grandparent(s), have no other children in the household, have father respondents, and have full-time working mothers (Table 1).

The mean score for caregiver cooking skills was 5.5 points among all participants (Table 2). The mean caregiver cooking skills score was lower for fathers (5.0 points) than for mothers (5.5 points). For each item in the cooking skills scale, mother scored higher than father, except for ‘like to cook’ (item 5). Compared with mothers, other respondents gave higher scores for the item of ‘like to cook’. Among the four cooking methods, fathers and other respondents rated stewing as more difficult than peeling, while mothers rated all methods as being of similar difficulty, although they had significant differences among the four methods (Table 2). ‘Like to cook’ was correlated with other cooking methods (*r* = 0.20–0.26, *p* < 0.0001), and especially highly correlated among fathers (*r* = 0.47–0.51, *p* < 0.0001) (Appendix A).

A low level of caregiver cooking skills was associated with low frequency of home cooking and child low frequency of vegetable intake (Table 3). After adjusting for potential confounders, caregivers with low-level cooking skills were 4.31 (95% CI: 2.68–6.94) times more likely to have lower frequency of home cooking than those with high level of cooking skills. Children with low level of caregiver cooking skills were 2.81 (95% CI: 2.06–3.84) times more likely to have low frequency of vegetable intake. Low level of caregiver cooking skills was not significantly associated with child breakfast skipping (AOR = 1.61, 95% CI: 0.97–3.53).

The models were adjusted for cohabitation status, siblings, respondent, mother’s age, and mother’s employment status.

A U-shaped association was found between caregiver cooking skills and child weight status (Table 4). Children with low level of caregiver cooking skills were 1.74 (95% CI: 1.08–2.82) times more likely to be obese and 1.84 (95% CI: 0.88–3.83) times more likely to be underweight, although the association with underweight status was not statistically significant. The mediation analysis showed that 91% of the association between low level of caregiver cooking skills and child obesity was mediated by frequency of home cooking.

The models were adjusted for cohabitation status, siblings, respondent, mother’s age, and mother’s employment status.

## 4. Discussion

To the best of our knowledge, this is the first study to investigate the associations between caregiver cooking skills and weight status of school children. Using a modified version of the existing cooking skills scale for use in the Japanese population, we found that a low level of caregiver cooking skills was associated with low frequency of home cooking and low frequency of vegetable intake in the child. Regarding child weight status, a U-shaped relationship was observed and a significant association was found between a low level of caregiver cooking skills and child obesity.

A low level of caregiver cooking skills was positively associated with child obesity and most of this association was explained by the frequency of home cooking. These findings are consistent with a previous study showing that infrequent home cooking was associated with child obese status [12]. A systematic review confirmed dietary benefits of home cooking, including greater consumption of healthier food groups, although most of the included studies involved adults [10]. In a study on child diets, home cooking was associated with higher vegetable consumption among children in the United Kingdom [26]. Consistent with that study, we also found that low level of caregiver cooking skills was associated with child low frequency of vegetable consumption.

A U-shaped relationship was observed, which indicates that low level of caregiver cooking skills also tended to be associated with child underweight status. This finding is consistent with a study on older Japanese adults showing that low level of cooking skills can lead to under-nutrition [19]. Children have difficulty preparing their own meals, and therefore their meals depend on their caregivers. Thus, low cooking frequency arising from low level of caregiver cooking skills may mean that children skip meals or eat low-energy diets, leading to them becoming underweight. To test this hypothesis, future studies are warranted to conduct more detailed dietary surveys on the frequency of children eating out.

A low level of caregiver cooking skills was associated with low frequency of home cooking. This finding is consistent with the previous study among older Japanese adults [19]. However, this result may be underestimated because there may have been more than one person in charge of cooking at home, such as the mother, father, grandmother, and older siblings, or a person different from the respondent may be the main cook. Furthermore, low level of caregiver cooking skills was not significantly associated with child breakfast skipping. One possible reason is that breakfast is generally a simple meal in Japan [27], and thus does not require a high level of cooking skills. Otherwise, it may be due to the child’s lack of time or appetite.

The validity of the cooking skills scale needs careful consideration. In the present study, mothers who tended to prepare food scored significantly higher on the cooking skills scale than fathers who were less likely to prepare food. This suggests that the modified version of the cooking skills scale used in the study had notable discriminant validity. The observed sex difference is consistent with previous findings on confidence regarding cooking skills, in which women were found to be more confident in their cooking skills than men [15,28,29]. We included four basic cooking methods in the cooking skills scale. Consistent with the previous study among older Japanese adults [19], fathers rated stewing as more difficult than peeling and boiling. Compared with the results for the Japanese older adults in the previous study, middle-aged women (mothers in the present study) scored almost the same as older women, while middle-aged men (fathers in the present study) tended to score higher than older men [19]. This generation difference among men may be explained by opportunities to learn cooking skills in school. In Japan, cooking education in schools for men started in 1947 and became compulsory in 1989 [30]. Therefore, older men had less opportunity to learn cooking in school.

We confirmed that ‘like to cook’ was correlated with other cooking methods, and especially highly correlated among fathers. This is plausible because women need to cook regardless of whether they like it because of the social norm [10], and as a result, their cooking skills will improve. We further confirmed that ‘like to cook’ was important for prevent low frequency of home cooking. As a result of analyzing the association between the single item ‘like to cook’ and the frequency of home cooking, caregivers with low level of liking to cook (score ≤ 4.0) were 2.21 (95% CI: 1.52–3.19) times more likely to have lower frequency of home cooking than those with high level of liking to cook (score > 4.0) after adjusting for potential confounders (data not shown). Given the importance of liking to cook for maintenance of a high level of cooking skills during the life course [23], it may be critical to examine subjects for liking to cook when examining the associations of cooking with diet-related outcomes.

There are some limitations to the present study. First, child frequency of vegetable dish intake was assessed using a single simple item. Future studies should use more detailed validated questions to assess which food groups and nutrients are associated with cooking skills. Second, we were only able to evaluate a limited number of child eating behaviors. Future studies are warranted to investigate the relationships between caregiver cooking skills and other aspects of child diets, such as amounts of energy and foods other than vegetables consumed, to understand the mechanisms. Third, we observed a ceiling effect for caregiver cooking skills, especially among mothers, similar to the findings in previous studies using the original cooking skills scale [15,19]. Given that mothers are often working, it may be useful to investigate not only their cooking methods (such as stewing), but also their ability to cook well in a short amount of time. In addition, more comprehensive validated measures are now available for assessing confidence in food and cooking skills in United Kingdom populations [31]. Therefore, it may be possible to use these measurement methods in the future. Fourth, the generalizability of the results may be low because our sample of school children was located in only one city in Japan. Fifth, we lacked data on some potentially confounding factors, such as caregivers’ nutritional knowledge and food preference. There may be a much more dynamic association between child obesity, caregiver cooking skills, and their liking for cooking. Finally, we were unable to assess causality because this was a cross-sectional study. However, in a previous study that examined the acquisition of cooking skills, more than half of the respondents reported that they had learned most of their cooking skills when they were teenagers and that these cooking skills were mainly taught by their mothers [32]. Randomized controlled trials in the younger generation before having children are needed in the future to clarify the effectiveness of caregiver’s ability to prepare meals for preventing obesity in children.

We found that a low level of caregiver cooking skills was associated with low frequency of home cooking, low frequency of child vegetable intake, and child obese status. Most of the association between low level of caregiver cooking skills and child obesity was mediated by the frequency of home cooking. The present findings are important for preventing unhealthy eating behaviors and obesity because COVID-19 is increasing the demand for home cooking. In addition, poor caregiver cooking skills can cause not only obesity in children but also less opportunity to learn cooking skills from caregivers, which may have an impact on the next generation (i.e., grandchildren of the current parents) due to the poor cooking skills of the children when they become parents [32]. In the future, it is necessary to clarify the causal relationships and promote research on support to improve caregiver cooking skills.

## Figures and Tables

**Table 1 nutrients-13-04549-t001:** Characteristics of Japanese school children and their caregivers (*n* = 5257).

	Total		Caregiver’s Cooking Skill		
	*n*	%	High	Low	*p*-Value ^a^
			*n* = 5010	*n* = 247	
			%	%	
Child’s status					
Sex					
Boy	2643	50.3	50.2	51.0	0.81
Girl	2614	49.7	49.8	49.0	
Age (year)					
9	1720	32.7	32.9	28.7	0.36
10	2479	47.2	46.9	51.4	
11	216	4.1	4.1	4.0	
12	286	5.4	5.4	6.9	
13	229	4.4	4.5	2.4	
14	327	6.2	6.2	6.5	
Dietary behaviors					
Frequency of vegetable intake					
Twice/day	2141	40.8	41.3	29.2	<0.001
Once/day	2549	48.5	48.8	41.7	
Less than 3 times/weeks (low frequency)	567	10.8	9.8	29.2	
Frequency of breakfast intake					
Everyday	4686	89.1	89.2	87.4	0.06
Often	436	8.3	8.3	7.7	
Rarely/never (breakfast skipping)	135	2.5	2.4	4.8	
Body weight status (BMI for age z score)					
Underweight (<−2SD)	123	2.3	2.3	3.6	0.02
Mild underweight (−2SD-<−1SD)	772	14.7	14.7	15.4	
Normal (−1SD-<+1SD)	3356	63.8	64.2	55.9	
Overweight (+1SD-<+2SD)	716	13.6	13.5	15.8	
Obesity(≥+2SD)	290	5.5	5.3	9.3	
Household status					
Cohabitation status					
Parents	4143	78.8	79.2	71.3	<0.001
Parents and grandparent (s)	383	7.3	7.2	8.1	
Single parent and grandparent (s)	610	11.6	11.5	13.8	
Single parent	65	1.2	1.1	3.6	
Other	56	1.1	1.0	3.2	
Other children in the household					
No	1017	19.3	18.8	29.6	<0.001
Yes	4240	80.7	81.2	70.4	
Household income (million yen)					
<3.00	559	10.6	10.7	10.1	0.20
3.00–5.99	1575	30.0	29.7	35.2	
6.00–9.99	1756	33.4	33.6	29.6	
≥10.0	591	11.2	11.4	8.5	
Missing	776	14.8	14.7	16.6	
Caregiver’s status					
Respondent					
Mother	4768	90.7	92.2	60.7	<0.001
Father	414	7.9	6.5	36.4	
Other	75	1.4	1.4	2.8	
Mother’s age (years)					
<35	499	9.5	9.6	7.7	<0.001
35–44	3074	58.5	58.4	60.7	
≥45	1512	28.8	29.0	23.9	
Missing	172	3.3	3.1	7.7	
Father’s age (years)					
<35	222	4.2	4.2	4.5	0.87
35–44	2356	44.8	44.7	47.0	
≥45	2014	38.3	38.4	37.2	
Missing	665	12.6	12.7	11.3	
Mother’s employment status					
Full-time	1114	21.2	21.0	24.7	0.002
Part-time	2516	47.9	48.2	41.7	
Self-employed	291	5.5	5.6	4.5	
Side work	115	2.2	2.2	2.0	
Not employed	1086	20.7	20.7	20.6	
Other/missing	135	2.6	2.4	6.5	
Mother’s BMI					
Underweight (BMI < 18.5)	596	11.3	11.4	10.5	0.06
Normal (18.5 ≤ BMI < 25.0)	3468	66.0	66.3	59.1	
Overweight (25.0 ≤ BMI < 30.0)	529	10.1	9.9	13.0	
Obesity (BMI ≥ 30)	99	1.9	1.8	2.8	
Missing	565	10.7	10.6	14.6	
Father’s BMI					
Underweight (BMI < 18.5)	77	1.5	1.4	2.0	0.27
Normal (18.5 ≤ BMI < 25.0)	2793	53.1	53.2	51.0	
Overweight (25.0 ≤ BMI < 30.0)	1125	21.4	21.2	26.3	
Obesity (BMI ≥ 30)	209	4.0	4.0	4.0	
Missing	1053	20.0	20.2	16.6	
Frequency of home cooking					
≥6 days/week	4552	86.6	87.8	61.9	<0.001
4–5 days/week	570	10.8	10.1	25.9	
≤3 days/week (low frequency)	135	2.6	2.1	12.1	

BMI: body mass index. ^a^ Differences were analyzed using Pearson’s chi-square test.

**Table 2 nutrients-13-04549-t002:** Cooking skills scale scores of Japanese caregivers (*n* = 5257).

Items	All				Respondent													
					Mother				Father					Other				
	*n* = 5257				*n* = 4768				*n* = 414					*n* = 75				
	Mean	SD	Coefficient	*p*-Value	Mean	SD	Coefficient	*p*-Value	Mean	SD		Coefficient	*p*-Value	Mean	SD		Coefficient	*p*-Value
Able to peel fruits and vegetables	5.78	0.69	reference		5.82	0.61	reference		5.32	1.20	***	reference		5.75	0.77		reference	
Able to make stir-fried meat and vegetables	5.81	0.67	0.03	0.03	5.85	0.56	0.04	0.007	5.31	1.28	***	−0.01	0.88	5.60	1.09	**	−0.15	0.24
Able to make miso soup	5.78	0.75	0.01	0.61	5.85	0.60	0.03	0.03	5.11	1.50	***	−0.21	0.001	5.49	1.13	***	−0.25	0.04
Able to make stewed dishes	5.64	0.95	−0.14	<0.001	5.73	0.77	−0.09	<0.001	4.59	1.79	***	−0.73	<0.001	5.36	1.36	**	−0.39	<0.001
Like to cook	4.45	1.32	−1.33	<0.001	4.44	1.30	−1.38	<0.001	4.49	1.46		−0.83	<0.001	4.96	1.18	**	−0.79	<0.001
Cooking skill scale	5.49	0.66			5.54	0.57			4.96	1.20	***			5.43	0.89			

Multiple comparisons between items on the cooking skills scale were analyzed using a mixed linear model procedure. Participant identification code was included as a random effect. Multiple comparisons between respondents were analyzed using Dunnett’s pairwise comparison method. ** *p* < 0.01, *** *p* < 0.001, versus mother.

**Table 3 nutrients-13-04549-t003:** Adjusted odds ratios of low frequency of home cooking, child low frequency of vegetable intake, and child breakfast skipping according to levels of caregiver cooking skills (*n* = 5257).

	Caregiver’s Low Frequency of Home Cooking	Child’s Low Frequency of Vegetable Intake	Child’s Breakfast Skipping
	AOR (95% CI)	AOR (95% CI)	AOR (95% CI)
Caregiver’s cooking skill			
High	ref	ref	ref
Low	4.31 (2.68–6.94)	2.81 (2.06–3.84)	1.61 (0.97–3.53)

AOR, adjusted odds ratio; CI, confidence interval.

**Table 4 nutrients-13-04549-t004:** Adjusted relative risk ratios of child obese, overweight, mild-underweight, and underweight status according to levels of caregiver cooking skills (*n* = 5257).

	Child’s Weight Status (Reference = Normal Weight (−1SD-<+1SD))			
	Obesity (≥+2SD)	Overweight (+1SD-< +2SD)	Mild Underweight (−2SD-< −1SD)	Underweight (<−2SD)
	ARRR (95% CI)	ARRR (95% CI)	ARRR (95% CI)	ARRR (95% CI)
Caregiver’s cooking skill				
High	ref	ref	ref	ref
Low	1.74 (1.08–2.82)	1.24 (0.85–1.82)	1.26 (0.86–1.84)	1.84 (0.88–3.83)

ARRR, adjusted relative risk ratio; CI, confidence interval.

## Data Availability

The data presented in this study are available on request from the corresponding author.

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
