# Peer review of "Associations of Caregiver Cooking Skills with Child Dietary Behaviors and Weight Status: Results from the A-CHILD Study"

_nutrients, 2021, doi:10.3390/nu13124549_

Round 1

Reviewer 1 Report

Accept with minor revisions:

  1. On line 80, please add 'with' between linked and health
  2. On line 105, what were the response options for the vegetable question?
  3. On page 6, in table 1, it should be 'Frequency of Home Cooking' instead of 'Frequency oh Home Cooking'.

Author Response

December 8, 2021

Response to Reviewer

Re: nutrients-1410722, Associations of caregiver cooking skills with child dietary behaviors and weight status: Results from the A-CHILD study

We thank the reviewer for their helpful comments. We have revised the manuscript by addressing the comment as below.

  1. On line 80, please add 'with' between linked and health

Reply:

Thank you for this useful suggestion. We have added 'with' to the Method section.

“Of these, 5,382 pairs (4,290 fourth-grade, 514 sixth-grade, and 578 eighth-grade) provided informed consent, returned all questionnaires, and could be linked with health checkup data (consent rate: 93%).”

  1. On line 105, what were the response options for the vegetable question?

Reply:

We thank you for this important comment and apologize for insufficiently addressing this point in the original manuscript. We have added the following sentence to the Method section

The three response items were: ‘twice/day’, ‘once/day’, and ‘< 3 times/week’.

  1. On page 6, in table 1, it should be 'Frequency of Home Cooking' instead of 'Frequency oh Home Cooking'.

Reply:

We thank you for this helpful comment. Accordingly, we have changed to 'Frequency of Home Cooking' in Table 1.

Again, thank you for these very kind and helpful comment. We hope that the paper is now suitable for publication in Nutrients.

Reviewer 2 Report

This paper describes a descriptive analysis of the association between frequency of home cooking, vegetable and breakfast intake, caregiver cooking skills and weight status of a large sample of Japanese, mainly 4th grade, children. In addition to a large sample size, the study used established assessments and directly measured children’s height and weight. The written style is clear and easy to read.

The main problem with the paper as it stands is the implicit direction of association assumed between caregiver cooking skills and children’s overweight/obesity. The authors acknowledge in the discussion that the study cannot address causality, it being a cross-sectional study.  This does not, however, stop them from making assumptions about the direction of association in the introduction (lines 47-54) and discussion (in particular regarding interventions, lines 226-228, 290-293). A much more dynamic association between child obesity, caregiver cooking skills, and their liking for cooking (and by implication choice of foods prepared outside the home) is likely. This means that simple interventions such as proposed (improving caregiver cooking skills) are unlikely to impact strongly on children’s future weight status.

Two minor specific issues:

  1. Please check lines 262-265 for meaning.
  2. It would be helpful to be more specific about the types of longitudinal studies that might develop understanding in this area (lines 283-285). Longitudinal research per se cannot establish causality.

Author Response

December 8, 2021

Response to Reviewer

Re: nutrients-1410722, Associations of caregiver cooking skills with child dietary behaviors and weight status: Results from the A-CHILD study

We thank the reviewer for their helpful comments. We have revised the manuscript by addressing the comment as below.

  1. The main problem with the paper as it stands is the implicit direction of association assumed between caregiver cooking skills and children’s overweight/obesity. The authors acknowledge in the discussion that the study cannot address causality, it being a cross-sectional study.  This does not, however, stop them from making assumptions about the direction of association in the introduction (lines 47-54) and discussion (in particular regarding interventions, lines 226-228, 290-293). A much more dynamic association between child obesity, caregiver cooking skills, and their liking for cooking (and by implication choice of foods prepared outside the home) is likely. This means that simple interventions such as proposed (improving caregiver cooking skills) are unlikely to impact strongly on children’s future weight status.

Reply:

Thank you for this very important comment. We have deleted the sentences in lines 226-228 and added and revised the sentences as follows:

Fifth, we lacked data on some potentially confounding factors, such as caregivers’ nutritional knowledge and food preference. There may be a much more dynamic association between child obesity, caregiver cooking skills, and their liking for cooking. Finally, we were unable to assess causality because this was a cross-sectional study. However, in a previous study that examined the acquisition of cooking skills, more than half of the respondents reported that they had learned most of their cooking skills when they were teenagers and that these cooking skills were mainly taught by their mothers.[33] Randomized controlled trials in the younger generation before having children are needed in the future to clarify the effectiveness of caregiver’s ability to prepare meals for preventing obesity in children.”

  1. Please check lines 262-265 for meaning.

Reply:

We apologize for not mentioning program. We have added the following sentence to the Discussion section.

As a result of analyzing the association between the single item ‘like to cook’ and the frequency of home cooking, caregivers with low level of liking to cook (score ≤4.0) were 2.21 (95% CI: 1.52–3.19) times more likely to have lower frequency of home cooking than those with high level of liking to cook (score >4.0) after adjusting for potential confounders (data not shown).”

  1. It would be helpful to be more specific about the types of longitudinal studies that might develop understanding in this area (lines 283-285). Longitudinal research per se cannot establish causality.

Reply:

Thank you for this useful suggestion. Accordingly, we have revised this sentence as follows:

Randomized controlled trials in the younger generation before having children are needed in the future to clarify the effectiveness of caregiver’s ability to prepare meals for preventing obesity in children.”

Again, thank you for these very kind and helpful comment. We hope that the paper is now suitable for publication in Nutrients.